# Assessment of PSIM (Prostatic Systemic Inflammatory Markers) Score in Predicting Pathologic Features at Robotic Radical Prostatectomy in Patients with Low-Risk Prostate Cancer Who Met the Inclusion Criteria for Active Surveillance

**DOI:** 10.3390/diagnostics11020355

**Published:** 2021-02-20

**Authors:** Matteo Ferro, Gennaro Musi, Deliu Victor Matei, Alessandro Francesco Mistretta, Stefano Luzzago, Gabriele Cozzi, Roberto Bianchi, Ettore Di Trapani, Antonio Cioffi, Giuseppe Lucarelli, Gian Maria Busetto, Francesco Del Giudice, Giorgio Ivan Russo, Marina Di Mauro, Angelo Porreca, Giuseppe Renne, Michele Catellani, Danilo Bottero, Antonio Brescia, Giovanni Cordima, Ottavio de Cobelli

**Affiliations:** 1Department of Urology, European Institute of Oncology, IRCCS, 20141 Milan, Italy; gennaro.musi@ieo.it (G.M.); d.v.matei@gmail.com (D.V.M.); francescoalessandro.mistretta@ieo.it (A.F.M.); stefano.luzzago@ieo.it (S.L.); gabriele.cozzi@ieo.it (G.C.); roberto.bianchi@ieo.it (R.B.); ettore.ditrapani@ieo.it (E.D.T.); antonio.cioffi@ieo.it (A.C.); michele.catellani@ieo.it (M.C.); danilo.bottero@ieo.it (D.B.); antonio.brescia@ieo.it (A.B.); giovanni.cordima@ieo.it (G.C.); ottavio.decobelli@ieo.it (O.d.C.); 2Department of Emergency and Organ Transplantation, Urology, Andrology and Kidney Transplantation Unit, University of Bari, 70124 Bari, Italy; giuseppe.lucarelli@inwind.it; 3Department of Urology and Renal Transplantation, University of Foggia Policlinico Riuniti, 71122 Foggia, Italy; gianmaria.busetto@unifg.it; 4Department of Urology, Sapienza Rome University Policlinico Umberto I, 00185 Rome, Italy; francesco.delgiudice@uniroma1.it; 5Department of Urology, University of Catania, 95123 Catania, Italy; giorgioivan.russo@unict.it (G.I.R.); marina.dimauro@unict.it (M.D.M.); 6Department of Urology, Veneto Institute of Oncology, 31033 Padua, Italy; angeloporreca@gmail.com; 7Department of Pathology, European Institute of Oncology, IRCCS, 20141 Milan, Italy; giuseppe.renne@ieo.it; 8Department of Oncology and Hemato-Oncology, University of Milan, 20122 Milan, Italy

**Keywords:** prostate cancer, neutrophil to lymphocyte ratio, platelet to lymphocyte ratio, lymphocyte to monocyte ratio, prognosis, active surveillance

## Abstract

Background: circulating levels of lymphocytes, platelets and neutrophils have been identified as factors related to unfavorable clinical outcome for many solid tumors. The aim of this cohort study is to evaluate and validate the use of the Prostatic Systemic Inflammatory Markers (PSIM) score in predicting and improving the detection of clinically significant prostate cancer (csPCa) in men undergoing robotic radical prostatectomy for low-risk prostate cancer who met the inclusion criteria for active surveillance. Methods: we reviewed the medical records of 260 patients who fulfilled the inclusion criteria for active surveillance. We performed a head-to-head comparison between the histological findings of specimens after radical prostatectomy (RP) and prostate biopsies. The PSIM score was calculated on the basis of positivity according to cutoffs (neutrophil-to-lymphocyte ratio (NLR) 2.0, platelets-to-lymphocyte ratio (PLR) 118 and monocyte-to-lymphocyte-ratio (MLR) 5.0), with 1 point assigned for each value exceeding the specified threshold and then summed, yielding a final score ranging from 0 to 3. Results: median NLR was 2.07, median PLR was 114.83, median MLR was 3.69. Conclusion: we found a significantly increase in the rate of pathological International Society of Urological Pathology (ISUP) ≥ 2 with the increase of PSIM. At the multivariate logistic regression analysis adjusted for age, prostate specific antigen (PSA), PSA density, prostate volume and PSIM, the latter was found the sole independent prognostic variable influencing probability of adverse pathology.

## 1. Introduction

Prostate cancer (PCa) is the second most commonly diagnosed neoplasm in men worldwide, with the highest incidence in the Western countries largely due to the widespread use of prostate-specific antigen (PSA) testing [1]. Radical prostatectomy (RP) is a therapeutic option for patients with clinically localized PCa, with a life expectancy of ≥10 years without serious comorbid conditions. [2,3]. However, many men with low-risk localized PCa may not benefit from active treatment modalities [4]. Recent observational and randomized clinical trials have shown that men harboring low-volume, low-grade PCa presented less than 6% risk of disease progression within a decade from diagnosis and demonstrated that cancer-related mortality from untreated patients with International Society of Urological Pathology (ISUP) grade 1–2 might be as low as 7% at 15-year follow-up [5]. Consequently, about 45% out of cT1c of detected PCa can be followed with a strict and careful surveillance program. Active surveillance (AS) is therefore mainly applicable to men with seemingly indolent cancer with the goal to defer or avoid treatment and its potential side effects, and to preserve a clinical window in which to intervene in cases of cancer progression. Consequently, the selection criteria and the decision to intensify treatment via curative options are suggested by indicators of potentially life-threatening disease [6]. Unfortunately, a growing body of evidence has demonstrated that these clinical variables may fail to accurately predict PCa aggressiveness and underestimate Gleason score (GS) with prostatectomy specimens in up to 66% of patients, therefore missing discrimination among indolent and clinically significant PCa (iPCa; csPCa) [7,8].

The ProtecT study [9] randomized men with localized prostate cancer to AS, RP or radiotherapy (RT). The study concluded that there are no significant differences in the primary outcome of PCa mortality at a median follow-up of 10 years. However, in the active surveillance group, higher rates of disease progression and metastasis were observed. 

To increase the prediction of each preoperative variable alone or combined in validated predictive nomograms [10], several risk stratification biomarkers were developed over the past years to this specific aim. Decipher and Oncotype Dx, which are tissue-based genomic classifiers based on RNA and specific cluster of cancer-genes, respectively, were developed as risk predictors of adverse pathology at RP in patients diagnosed with low or intermediate disease on biopsy [11,12]. However, their adoption into clinical practice has been modest due to the absence of long-term, prospective investigations, and cost utility and effectiveness studies, all of which could elucidate economic benefits and the overall net benefit of genetic markers on patient survival outcomes [13]. 

On the other hand, systemic markers of inflammation have been previously associated with poor prognosis for many solid tumors [14,15,16]. The neutrophil-to-lymphocyte ratio (NLR) along with monocyte-to-lymphocyte ratio (MLR), platelets-to-lymphocyte ratio (PLR), and eosinophil-to-lymphocyte ratio (ELR) have been proposed also in the contest of PCa [17]. Recent implementation of a systemic inflammatory marker model (SIM score) based on the measurement of the aforementioned inflammatory markers has shown some success in predicting bladder cancer (BCa) recurrence and progression rates [4]. Therefore, the aim of the present cohort study is to assess and validate the use of prostatic systemic inflammatory markers (PSIM) score in predicting and improving the detection of significant disease at robotic radical prostatectomy in patients with low-risk prostate cancer who met the inclusion criteria for active surveillance.

## 2. Material and Methods

We reviewed the medical records of 260 patients who fulfilled the inclusion criteria for “Prostate Cancer Research International: Active Surveillance” [18] defined as follows: clinical stage T2a or less, PSA level < 10 ng/mL, 2 or fewer cores involved with cancer after a biopsy scheme of at least 12 cores, Gleason score (GS) ≤ 6, and PSA density (PSA-D) < 0.2 ng/mL/cc. We compared the pathological findings between specimens after RP and prostate biopsies. RP specimens were processed and evaluated according to the Stanford protocol [19] by a single, experienced, genitourinary pathologist (G.R.) blinded to index tests results. PCa was identified and graded according to the 2002 American Joint Committee on Cancer staging system and the ISUP grade group (GG) in accordance to ISUP 2014 consensus conference [20]. 

None of the patients received neoadjuvant androgen-deprivation therapy or 5-ARI (5-alpha reductase inhibitors) or had a history of prostate surgery.

Thirty days before the surgery we collected the laboratory data and the values of NLR, PLR and MRL. Patients with acute or chronic infection or hematologic disorders were excluded from the study.

### Statistical Analysis 

Continuous variables are presented as the median and interquartile range (IQR). Differences between groups were assessed using a Kruskall–Wallis or Mann–Whitney U test as appropriate. Categorical variables were tested using an χ² test or Fisher’s exact test. 

Multivariable logistic regression analysis (MVA) was performed to identify factors predictive of extracapsular extension (ECE) and ISUP ≥ 2 using the variables measured. 

NLR, PLR and MLR were dichotomized according to the following cutoffs: 2.0, 118, and 5.0, respectively [1,2,3]. The PSIM score was calculated by assigning 1 point when the value exceeded the specified threshold (NLR 2.0, PLR 118 and MLR 5.0), and then we added the points to obtain a final score between 0 and 3 [4]. 

All statistical analyses were carried out using Stata v14 (StataCorp, College Station, TX, U.S.). For all comparisons, the significance level was set to *p* < 0.05 for differences among groups. 

## 3. Results

Table 1 lists the baseline characteristics of the study cohort. Median age was 62.0 (61.0–63.2), median NLR was 2.07 (1.67–2.66), median PLR was 114.83 (93.89–136.3), median MLR was 3.69 (3.03–4.43), and median PSIM was 1.0 (IQR: 1.0–1.0). When dichotomizing patients according to defined cut-offs, 155 (59.61%), 113 (43.46%) and 39 (15%) had, respectively, an NLR ≥2.0, PLR ≥118 and MLR ≥ 5.0.

Table 2 shows the clinical and pathological characteristics according to the previously defined markers’ cut-offs. As concerning pathological variables, we found a statistically significant different increase rate of pathological ISUP ≥ 2 in those patients with NLR ≥ 2.0 (47.1% vs. 20%; *p* = 0.001) and with MLR ≥ 5.0 (43.6% vs. 34.8%; *p* = 0.001). In total, 96 (37.2%) had a PSIM ≥ 2. 

Table 3 shows the clinical and pathological variables according to the PSIM. We found a significant increase in rate of pathological ISUP ≥ 2 with the increase in PSIM from 0 to 3 (14.8%, 33.6%, 48.3% and 100%; *p* = 0.01). 

At multivariate analysis (MVA) adjusted for age, prostate volume, PSA and PSA density, there were no significant differences. On the contrary, PSIM was the sole independent prognostic variable influencing probability of adverse pathology at RP (OR: 2.17, [95%CI: 1.33–3.54]; *p* = 0.001). MVA is shown on Table 4.

## 4. Discussion

About 23 to 42% of all U.S. screen detected PCa has been assumed to be overtreated, according to the detection and progression estimate model described by Draisma et al [21]. Out of these, PSA detection was responsible for up to 12.3 years of lead-time bias [22]. For this reason, Epstein et al [6] introduced clinical criteria to predict pathologically “insignificant” PCa. These variables and their integration in predictive nomograms have been considered as the selection criteria for patients enrolled in “deferred” treatment strategies such as watchful waiting (WW) or AS [10,23,24]. 

Long-term follow-up analyses assessing overall survival (OS) and cancer specific survival (CSS) for PCa patients on AS have reported excellent outcomes [25]. Nevertheless, different experiences have shown that as many as 8% of PCa qualified as insignificant was not organ-confined, based on postoperative findings, and more than one-third of these patients underwent reclassification over the follow-up period [7]. Most of these patients will eventually require curative treatment due to disease upgrading and/or upstaging in the future [26]. In this complex decision-making scenario, we have to acknowledge the existence of the considerable variations and heterogeneity in the literature with regard to AS patient eligibility, follow-up policies and reclassification criteria, and the lack of standardized tools aiding urologists in predicting outcomes for such a patient’s category [27]. In the present study, we assessed the diagnostic utility of a multivariable model based on a combination of three systemic inflammatory markers to predict the incidence of clinically significant PCa in low- and very low-risk patients eligible for AS. In a previous exploratory study from our group [28], we found that in patients without systemic or prostate-related inflammation, high NLR, PLR and ELR were significantly associated with upgrading, but not with upstaging. The rationale behind the potential utility of such inflammatory markers likely could be explained by a favorable immune microenvironment for PCa development and even possible subsequent metastasis. The high-volume production, release of cytokines and growth factors from tumor-infiltrating inflammatory cells might be able to promote angiogenesis, proliferation, migration and finally invasion, which has been previously demonstrated [29,30].

Eosinophils and platelets have shown to be promoters of cancer progression and metastasis via the secretion of several cytokines such as interleukin-6 (IL-6), tumor necrosis factor-α (TNF-α) (responsible for a subverted host response to inflammation-induced tumors) and through platelet β3 integrins, which are particularly active in the process of bone dissemination [31,32]. The role of NLR has been extensively studied and was shown to be a negative prognostic marker in different stages of PCa. Van Soest et al. [33] demonstrated the negative impact of elevated NLR levels in patients diagnosed with metastatic castration resistant PCa (mCRPC) resistant to multiple systemic therapies. In one of the first experiences directly assessing NLR within the prebiopsy setting, Gokce et al. [34] demonstrated how ISUP group 4–5 patients had a significantly higher mean NLR when compared to ISUP group 1 (3.64 vs. 2.54, *p* = 0.0001) and group 2–3 (3.64 vs. 2.58, *p* < 0.01). This observed trend, despite lacking reliable multivariable logistic regression models assessing the weight of different confounders, should be considered as the first piece of evidence depicting the existence of a higher inflammatory background that could potentially help in discriminating among GS group categories. 

A growing body of evidence has moved forward in implementing these inflammatory markers in more complex nomograms that would rely on the influence of prebiopsy imaging such as multiparametric magnetic resonance imaging (mpMRI) of the prostate [35,36]. In particular, Sun et al. [37] combined Prostate Imaging Reporting Data System (PI-RADS) and NLR to improve the detection of csPCa in men with PSA < 10 ng/mL at first biopsy. The authors found a better diagnostic prediction trend with the implementation of the full risk model (baseline variables + NLR + PI-RADS v2) vs. baseline + NLR alone (area under the curve [AUC]:0.854, 95%CI 0.807−0.900 vs. 0.813, 95%CI 0.762−0.865 respectively). Regarding association with biochemical parameters, as previously reported, the prostate health index (PHI) and inflammation score such as pretreatment NLR, may be associated with biochemical recurrence-free survival in patients undergoing radical prostatectomy [38,39].

With these exciting results as an important background, and with the aim to implement and utilize inflammatory markers to predict the likelihood of adverse prognosis in men undergoing prostate biopsy for rising PSA levels, we applied a novel PSIM score to better quantify the importance of the prebiopsy markers in the decision-making algorithm. The study was not without limitations as the authors did not demonstrate how NLR may be used to differentiate between benign associated prostate conditions such as benign prostatic hyperplasia (BPH) and prostatitis, which may limit the net benefit of the tool alone. 

Our work differs from the aforementioned studies for several reasons: First, the application of the PSIM score in this study was significantly associated with increased probability of ISUP group ≥ 2 both as a continuous or dichotomized variable (independently identified as preoperative clinical predictor of adverse pathology at MVA). Second, the cumulative effect of our prediction score was seen in a more contemporary series consisting of a large cohort of patients based on strict inclusion criteria. Finally, different from prior experiences, we utilized a widely inclusive statistical regression model to involve all possible clinical confounders and commonly adjusted for adverse pathology prediction at a definitive histological RP report. 

While our results are promising, our study is not without limitations. First, similar to previous analyses that reported outcomes from the use of inflammatory markers, our study is based on a retrospective design that has its own inherent limitations. Second, we did not perform a sensitivity analysis on the impact of different benign inflammatory conditions on the prediction of our assessed SIM score, therefore lacking in the ability to adjust the variability in our outcomes on a prebiopsy prostatic status. Finally, our SIM score as conceptualized and presented, and did not include other potential inflammatory markers or immune system variables that could have been used to further substantiate or invalidate the model.

## 5. Conclusions

Our single-center series demonstrated that patients with an increasing PSIM score before prostate biopsy are associated with higher probability of csPCa at the final RP pathology report. If on one hand these results could in future be potentially translated into clinical practice to better stratify patients presenting with low-risk PCa at prostate biopsy who might actually benefit from early active treatment instead of AS, on the other hand, future larger prospective and multi-institutional experiences are mandatory to validate the presented findings. Moreover, future risk stratification nomograms combining other critical covariates such as different inflammatory markers and imaging should be investigated to verify the possibility of an even better risk assessment. 

## Figures and Tables

**Table 1 diagnostics-11-00355-t001:** Baseline Characteristics of the Study Cohort.

Variables	*n* = 260
Age (years), median (95%CI)	62.0 (61.0–63.2)
PSA (ng/ml), median (95%CI)	5.6 (5.32–5.95)
PSA density (ng/ml/cc), median (95%CI)	0.12 (0.10–0.13)
Total Number of cores, median (95%CI)	12.0 (12.0–14.0)
Positive cores, n (%)	
1	137 (52.7)
2	123 (47.3)
Clinical stage, n (%)	
cT1c	231 (88.85)
cT2	29 (11.15)
Pathological stage, n (%)	
pT2	189 (72.69)
pT3a	68 (26.15)
pT3b	3 (1.15)
Pathological lymph node, n (%)	
N1	1 (0.38)
N0	38 (14.62)
Nx	221 (85.0)
Pathological Gleason Score, n (%)	
ISUP 1	166 (63.8)
ISUP 2	75 (28.8)
ISUP 3	17 (6.6)
ISUP 4	2 (0.8)
NLR, median (IQR)	2.07 (1.67–2.66)
PLR, median (IQR)	114.83 (93.89–136.3)
LMR, median (IQR)	3.69 (3.03–4.43)

95%CI = 95% confidence interval for median; NLR = neutrophil-to-lymphocyte ratio; PLR = platelets-to-lymphocyte ratio; LMR = lymphocyte-to-monocyte ratio; ISUP = International Society of Urological Pathology.

**Table 2 diagnostics-11-00355-t002:** Association of baseline clinicopathologic characteristics and NLR, PLR and LMR cut-off.

Variables	NLR < 2.0 (*N* = 105)	NLR ≥ 2.0 (*N* = 155)	*p*-Value	PLR < 118 (*N* = 147)	PLR ≥ 118 (*N* = 113)	*p*-Value	LMR < 5.0 (*N* = 221)	LMR ≥ 5.0 (*N* = 39)	*p*-Value
Age (years), median (95%CI)	61.0 (59.0–63.0)	63.0 (61.0–64.0)	0.13	62.0 (61.0–64.0)	63.0 (60–64)	0.62	63.0 (61.5–64.0)	60.0 (58.0–62.8)	0.01
PSA (ng/ml), median (95%CI)	5.5 (5.03–6.02)	5.7 (5.30–6.01)	0.31	5.6 (5.30–6.00)	5.5 (5.03–6.09)	0.45	5.7 (5.31–6.00)	5.5(4.54–6.29)	0.63
PSA density (ng/ml/cc), median (95%CI)	0.10 (0.10–0.12)	0.12 (0.11–0.14)	0.02	0.12 (0.10–0.13)	0.12 (0.10–0.13)	0.48	0.12 (0.10–0.13)	0.11 (0.09–0.15)	0.75
Prostate volume (mL), median(95%CI)	50.0 (45.0–53.0)	49.0 (46.0–52.0)	0.71	49.0 (46.0–50.7)	51.0 (45.0–55.0)	0.63	49.0 (46.0–50.4)	53.0 (38.2–60.0)	0.91
ECE, *n* (%)	28 (23.6)	43 (27.7)	0.01	37 (25.1)	34 (30)	0.71	54 (24.4)	17 (43.6)	0.001
Pathological ISUP ≥ 2, *n* (%)	21 (20)	73 (47.1)	0.001	44 (29.9)	50 (44.2)	0.40	77 (34.8)	17 (43.6)	0.001

95%CI = 95% confidence interval for median; NLR = neutrophil-to-lymphocyte ratio; PLR = platelet-to-lymphocyte ratio; LMR = lymphocyte-to-monocyte ratio; ECE = extracapsular extension; ISUP = International Society of Urological Pathology.

**Table 3 diagnostics-11-00355-t003:** Association of baseline clinicopathologic characteristics and number of cumulative marker score in the total cohort.

	Systemic Inflammatory Markers				
Variables	0 (*N* = 54)	1 (*N* = 110)	2 (*N* = 91)	3 (*N* = 5)	*p*-Value
Age (years), 95%CI	61.0 (59.0–63.5)	63.0 (61.0–64.0)	63.0 (60.0–65.0)	56.0 (46.0–66.0)	0.21
PSA (ng/ml), 95%CI	5.65 (4.97–6.62)	5.68 (5.39–6.08)	5.30 (4.92–5.92)	9.02 (6.09–9.40)	0.04
PSA density (ng/ml/cc), median (95%CI)	0.11 (0.10–0.13)	0.12 (0.10–0.14)	0.12 (0.10–0.13)	0.18 (0.11–0.19)	0.06
Prostate volume (ml), median (95%CI)	50.0 (43.4–52.5)	49.0 (45.0–52.0)	50.0 (45.0–56.0)	52.0 (32.0–60.0)	0.86
ECE, *n* (%)	11 (20.4)	30 (27.3)	26 (28.6)	4 (80.0)	0.03
Pathological ISUP ≥2, *n* (%)	8 (14.8)	37 (33.6)	44 (48.3)	5 (100.0)	0.001

95%CI = 95% confidence interval for median; ECE = extracapsular extension; ISUP = International Society of Urological Pathology.

**Table 4 diagnostics-11-00355-t004:** Logistic regression for unfavorable disease.

Variables	O.R.	95%CI	*p*-Value
Age	1.02	0.97–1.08	0.30
PSA	1.23	1.02–1.49	0.02
PSA density	1.51	1.10–1.65	0.01
Prostate volume	0.98	0.96–1.01	0.23
PSIMS	2.17	1.33–3.54	0.001

95%CI = 95% confidence interval for median; O.R. = odds ratio; PSIMS = Prostatic Systemic Inflammatory Markers Score.

## Data Availability

Data are conserved at European Institute of Oncology (IEO), Milan, Italy repository and are available for eventual consultation.

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
