# Peer review of "Assessment of PSIM (Prostatic Systemic Inflammatory Markers) Score in Predicting Pathologic Features at Robotic Radical Prostatectomy in Patients with Low-Risk Prostate Cancer Who Met the Inclusion Criteria for Active Surveillance"

_diagnostics, 2021, doi:10.3390/diagnostics11020355_

Round 1

Reviewer 1 Report

This study was reported the utility of PSIM in patients with low-risk prostate cancer who underwent RARP. The reviewer would like to suggest some critiques as follows.

Major revision

  1. The reviewer think that the Title is unclear. What is “pathological significant disease”?
  2. The authors should follow the journal style indicated in the instructions, especially the Reference sections.
  3. The authors should spell out csPCa, RP, NLR, PLR, LMR, and PSA at the Abstract.
  4. On page 2, line 33; ofPCa → of PCa
  5. On page 3, line 31, “At multivariate…..wasn’t found” is unclear. The reviewer think that “ On multivariate analysis, there were no significant differences …” is correct.

Author Response

Dear reviewer thank you for your work.

We modified the text as follow:

1) title has been changed

2) reference section has been adapted to journal style

3) in the abstract we added required abbreviations

4) corrected

5) corrected

Reviewer 2 Report

An interesting manuscript  showing an assessment of PSIM score in patients with low risk prostate cancer. In my opinion, It exist many scoring systems which were applied for estimate diagnostic tool in patients with cancerous ethiologies. I think that clearly depends on the patients group selection . Authors well documented use the PSIM score in patients with low risk prostate cancer, nevertheless, My question which authors need to discuss is  if the course will be applicable for for patients with pT0-PT1 stage  and if is in literature described some corellation with biochemical parmetres including PHI etc. After discussing and comment this minor suggestions, I approve the manuscript for publication.

Author Response

Dear reviewer, thank you for you work.

Regarding your request about applicability of our model to pT0-pT1 patients it wasn't used with these patient, even in literature there isn't anything at this regard. On the other side we added a sentence (in red) and two references regarding correlation with PHI and biochemical parameters.